# Impact of fluoroquinolone resistance on the cost-effectiveness of empiric treatment for multidrug- or rifampicin-resistant tuberculosis

Chaelin Kim[1]*, Sedona Sweeney[2], Hojoon Sohn[3,4], Gwenan M. Knight[1‡], C. Finn McQuaid[1‡]

1 Department of Infectious Disease Epidemiology and Dynamics, TB Modelling Group, Tuberculosis Centre, Centre for Mathematical Modelling of Infectious Diseases, Antimicrobial Resistance Centre, London School of Hygiene and Tropical Medicine, London, United Kingdom, 2 Department of Global Health and Development, London School of Hygiene & Tropical Medicine, London, United Kingdom, 3 Department of Preventive Medicine and Department of Human Systems Medicine, Seoul National University College of Medicine, Seoul, Republic of Korea, 4 Seoul National University Institute of Health Policy and Management, Seoul, Republic of Korea

‡ These authors are joint senior authors.
* chaelin.kim1@lshtm.ac.uk

## Abstract

The WHO recommends the bedaquiline, pretomanid, and linezolid (BPaL) regimen with the additional fluoroquinolone antibiotic moxifloxacin (BPaLM) for initial treatment of multidrug- or rifampicin-resistant tuberculosis (MDR/RR-TB). However, fluoroquinolone drug susceptibility testing (DST) coverage for MDR/RR-TB is only around 55% globally, and the efficacy of moxifloxacin may be compromised in settings with high fluoroquinolone resistance. We extended a previous Markov cohort model to assess the cost-effectiveness of the empirical use of BPaLM as a replacement for BPaL for the treatment of MDR/RR-TB in four high MDR/RR-TB burden countries. We obtained fluoroquinolone resistance rates in these countries from WHO surveillance data and parameterised treatment efficacy from recent trial results. We performed scenario analyses across varying fluoroquinolone resistance prevalence and performed Monte Carlo simulations to generate uncertainty intervals for our primary cost-effectiveness estimates. BPaLM incurred higher costs than BPaL but averted more disability-adjusted life years, with incremental cost-effectiveness ratios below 50% of each country's 2019 GDP per capita. This finding remained robust across a feasible fluoroquinolone resistance prevalence range (0–70%). In the absence of fluoroquinolone DST, empirical use of BPaLM resulted in $58 (interquartile range: $49-$73), $32 (IQR: $23-$53), $35 (IQR: $28 - $51), $174 (IQR: $161 - $209) per DALY averted in Georgia, India, the Philippines, and South Africa respectively. Our findings support the empirical use of BPaLM as a potential replacement for BPaL for the treatment of MDR/RR-TB in the absence of fluoroquinolone DST, even if fluoroquinolone resistance prevalence were to increase, reinforcing recent WHO recommendations.

**Data availability statement:** All key parameters and inputs used in this study are detailed within the paper and the Supporting information.

**Funding:** This study was funded by the UK Medical Research Council London Intercollegiate Doctoral Training Partnership Studentship (grant number MR/W006677/1) awarded to CK. The UK Medical Research Council London Intercollegiate Doctoral Training Partnership is a partnership between City, St George's University of London, and London School of Hygiene & Tropical Medicine (https://mrc-lid.lshtm.ac.uk/). The funder had no role in study design, data collection and analysis, decision to publish, or preparation of the manuscript.

**Competing interests:** The authors have declared that no competing interests exist.

## Introduction

Multidrug- or rifampicin-resistant tuberculosis (MDR/RR-TB) remains a major global health problem, with an estimated 400,000 new cases and 150,000 deaths in 2023 [1]. Appropriate treatment based on accurate drug susceptibility testing (DST) is directly linked to successful treatment outcomes [2]. However, the diagnosis and treatment of drug-resistant (DR-)TB continues to pose a significant challenge [3]. Limited test coverage, inadequate infrastructure, long delays in receiving results, and the unavailability of DST for certain TB drugs often prevent treatment initiation based on DST results [4]. This may necessitate empirical treatment, treating without DST results, or starting treatment empirically until DST results become available [1,5].

Updated World Health Organization (WHO) guidelines for MDR/RR-TB treatment recommend an all-oral 6-month regimen of bedaquiline, pretomanid, and linezolid with (BPaLM) or without (BPaL) moxifloxacin [6]. This guideline is based on the ZeNix [7,8] and TB-PRACTECAL [9–11] trials, demonstrating the superiority of all-oral 6-month regimens compared to traditional regimens with longer, often more complex treatment regimens that may have included injectable agents. The all-oral 6-month regimens showed improved treatment outcomes and reduced adverse events compared to these traditional regimens. The TB PRACTECAL trial revealed that the BPaLM regimen had higher treatment success and fewer treatment failures or recurrences, with only small differences in the occurrence of adverse events compared to both the BPaL (without moxifloxacin) and the BPaLC (with clofazimine substitution) regimens [10,11]. A model-based cost-effectiveness analysis found that all three regimens (BPaL, BPaLC, BPaLM) were also cost-saving compared with the traditional regimens in India, South Africa, the Philippines and Georgia. BPaLM was the preferred regimen in all four countries at a willingness-to-pay threshold of 50% of the primary cost-effectiveness threshold set at each country's 2019 GDP per capita per disability-adjusted life-year averted [12].

In an empirical treatment setting for detected MDR/RR-TB, where second-line drug susceptibility results for moxifloxacin are not available, guidelines suggest the use of BPaLM as the initial treatment for MDR/RR-TB, with a switch to the BPaL regimen without moxifloxacin if moxifloxacin resistance is later detected, potentially reducing side effects and costs [6]. However, access to DST for second-line TB drugs remains limited in many high-burden countries [1], is often more expensive than other TB diagnostics, and requires specialised laboratory infrastructure and expertise [13,14]. As a result, coverage for fluoroquinolone DST among MDR/RR-TB patients is only 55% globally, with lower coverage in low-resource settings [1].

Including a fluoroquinolone such as moxifloxacin in these regimens can improve outcomes, but its effectiveness is reduced if the infecting strain has fluoroquinolone resistance. However, pre-extensively drug-resistant TB (pre-XDR-TB), defined as MDR/RR-TB with additional resistance to fluoroquinolones, was found in 19% of MDR/RR-TB cases globally in 2023, with high setting-specific variance [1]. Among tested MDR/RR-TB cases, fluoroquinolone resistance rates ranged from high levels in India (37%) and Georgia (27%) to lower levels in South Africa (19%) and the Philippines (9%) in 2019 [15].

Without evidence comparing these regimens under empirical treatment conditions, clinicians must make treatment decisions based on incomplete information. Giving the BPaLM regimen to patients with pre-XDR-TB could result in the unnecessary use of moxifloxacin, increasing costs and potential adverse events with little added benefit [16]. Conversely, using the BPaL regimen without moxifloxacin for patients sensitive to fluoroquinolones may forgo the additional efficacy provided by this drug. Therefore, understanding the impact of varying fluoroquinolone resistance prevalence is crucial in determining the optimal empirical treatment strategy in different settings, where the potential benefits and harms must be balanced carefully [17]. Despite the clinical importance of these treatment decisions, there is currently limited evidence to guide empirical treatment choices between the new all-oral regimens.

To address this critical knowledge gap, we conducted the first economic evaluation directly comparing empirical BPaLM versus BPaL treatment across varying fluoroquinolone resistance scenarios. While previous cost-effectiveness analyses assumed DST availability or compared new regimens only against traditional treatments [12,17–19], our study directly compares these two novel short all-oral regimens under different fluoroquinolone resistance scenarios. This analysis addresses a critical evidence gap in empirical treatment settings where DST results are unavailable, providing essential guidance for clinical decision-making.

## Methods

We extended a previously developed Markov cohort model [12] to estimate the incremental cost-effectiveness ratios (ICERs) of empirically treating MDR/RR-TB and pre-XDR patients with BPaLM relative to BPaL when fluoroquinolone DST was not available and under different fluoroquinolone resistance levels. Our updated model retains the core structure while adding parameters to estimate treatment success for MDR/RR-TB and pre-XDR-TB populations in empirical treatment scenarios. We use the same four countries, Georgia, India, the Philippines, and South Africa, as in the previous analysis to reflect diverse population and epidemiological characteristics, including high TB burden and geographical representation, and the availability of cost data.

### Model description

We assumed an empirical treatment setting in which patients with MDR/RR-TB were either fluoroquinolone-susceptible or resistant but did not have access to the second-line drug DST. We developed two treatment scenarios for this population. In the first scenario, all patients receive the BPaL regimen without moxifloxacin, potentially foregoing additional treatment benefits for MDR/RR-TB patients. In the second scenario, all patients receive the BPaLM regimen with moxifloxacin, which may result in unnecessary drug use and costs for pre-XDR-TB patients (S1 Table).

Consistent with the previous model [12], we used a Markov model to simulate the progression of patients through different health states and treatment pathways over a 20-year time horizon with monthly cycles from a provider perspective (S1 Fig). The analysis followed the Consolidated Health Economic Evaluation Reporting Standards (CHEERS) 2022, as described in the supplementary materials (S1 Checklist) [20].

Our analysis extended the previous model by incorporating fluoroquinolone resistance-specific treatment success outcomes (S2 Text). The previous model estimated treatment success by multiplying country-specific traditional regimen treatment success by the ratio of treatment success probabilities between intervention and traditional regimen arms in the TB-PRACTECAL trial [11]. We used a similar approach, but generated composite risk ratios from the TB-PRACTECAL trial [11] that account for differential efficacy in populations with MDR/RR-TB and pre-XDR-TB, to account for different scenarios of fluoroquinolone resistance prevalence.

To estimate the true risk ratios specific to populations with MDR/RR-TB and pre-XDR-TB, we back-calculated from the baseline fluoroquinolone resistance prevalence (24.6%, denoted here as FQR) in the randomised BPaLM group of the trial [11] using equations [Equations 1 and 2]. The trial results provided risk ratios for BPaLM and BPaL compared to traditional regimens in a mixed population of MDR/RR-TB and pre-XDR-TB patients ($RR_{BPaLM\_PRACTECAL}$, $RR_{BPaL\_PRACTECAL}$

respectively). As the trial was not powered to analyse outcomes separately by fluoroquinolone resistance status, we used the overall proportion of patients without an unfavourable outcome at 108 weeks [11]. This proportion was used to calculate the actual risk ratio, which reflects the likelihood of achieving a favourable outcome with the new regimens (BPaLM or BPaL) compared to the traditional regimens. By assuming that BPaLM efficacy in the pre-XDR-TB population is equivalent to that of BPaL, given the limited role of moxifloxacin in treating TB in fluoroquinolone-resistant strains, we were able to derive the true population-specific risk ratios linked to the regimens ($RR_{BPaLM}$ and $RR_{BPaL}$):

$$RR_{BPaL\_PRACTECAL} = RR_{BPaL} \cdot (1 - FQR) + RR_{BPaL} \cdot FQR \qquad \text{[Equation 1]}$$

$$RR_{BPaLM\_PRACTECAL} = RR_{BPaLM} \cdot (1 - FQR) + RR_{BPaL} \cdot FQR \qquad \text{[Equation 2]}$$

Using these estimated fluoroquinolone-resistance-specific risk ratios for new regimens compared to traditional regimens, and WHO-reported data on the country-specific traditional regimen treatment outcomes, where the standard treatment success rate is denoted as traditional regimen success, and fluoroquinolone resistance prevalence, we calculated composite treatment success rates [Equations 3 and 4].

$$BPaL\ composite\ success\ rate = traditional\ regimen\ success \cdot [(1 - FQR) \cdot RR_{BPaL} + FQR \cdot RR_{BPaL}] \qquad \text{[Equation 3]}$$

$$BPaLM\ composite\ success\ rate = traditional\ regimen\ success \cdot [(1 - FQR) \cdot RR_{BPaLM} + FQR \cdot RR_{BPaL}] \qquad \text{[Equation 4]}$$

Based on these estimations, we assessed the cost-effectiveness of the empirical use of BPaLM as a potential replacement for BPaL, using the incremental cost per DALY averted and country-specific cost-effectiveness thresholds [21]. To appropriately incorporate the uncertainty in our estimates, we conducted a probabilistic sensitivity analysis (PSA) by varying input parameters such as treatment success rates, unit costs, DALY weights, and transition probabilities over 200 simulation runs, with parameter values, standard errors, and probability distributions specified in S2 Table. For the main analysis, we kept fluoroquinolone resistance prevalence fixed at baseline country-specific values (Fig 1).

## Data resources

We used country-specific treatment outcomes and data on fluoroquinolone resistance testing coverage and fluoroquinolone resistance prevalence from WHO surveillance and outcome reports (Table 1) [15]. For the remaining data on costs, DALY weights, and transition probabilities, we used country-specific and non-specific parameters drawn from multiple sources, including national MDR treatment guidelines, the Value TB dataset, published literature, and expert opinion (S2 Table). We incorporated adverse events using the same approach, with aggregate costs derived from TB-PRACTECAL trial incidence data, Global Burden of Disease weights for health impacts, literature-based cost estimates, and expert opinion. We have excluded any costs associated with second-line drug susceptibility testing, such as liquid culture medium drug susceptibility testing for second-line drugs or second-line drug line probe assay, as we focus on the empirical treatment situation where second-line drug susceptibility testing is not available.

## Sensitivity analyses

To examine the impact of fluoroquinolone resistance, we conducted a separate analysis in which we varied only the fluoroquinolone resistance prevalence, setting it to values of 1%, 50%, 70%, and 90%, while sampling all other parameters as in the main PSA to generate cost-effectiveness acceptability curves (Fig 2). To examine the impact of key parameters on ICER values, we conducted univariate sensitivity analyses by varying fluoroquinolone resistance

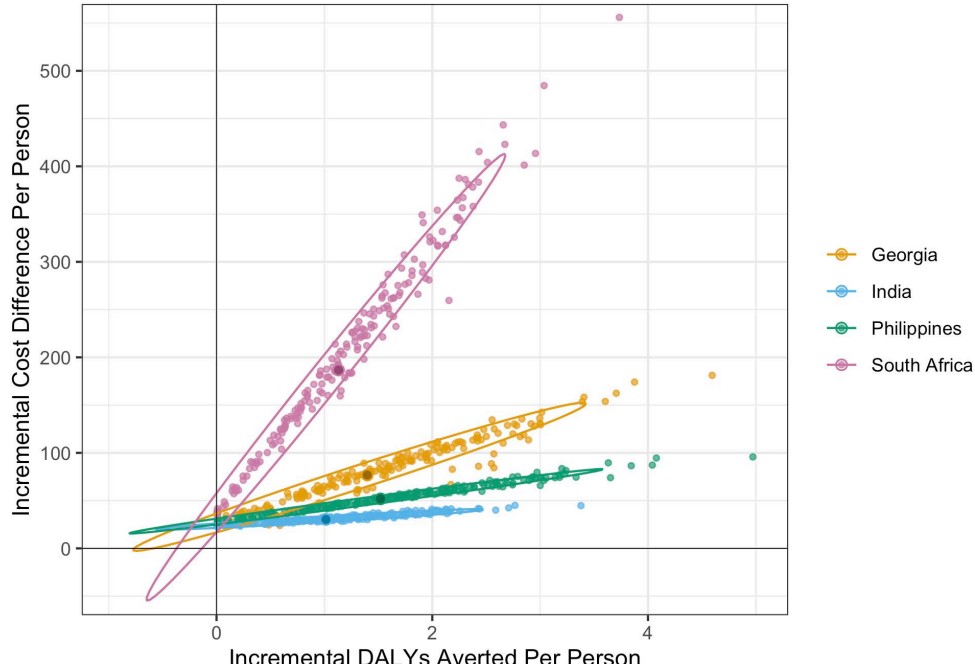

**Fig 1. Cost-effectiveness plane of ICER of using BPaLM as a replacement for BPaL.** This figure illustrates the ICER of using the BPaLM regimen as a replacement for the BPaL regimen for the empirical treatment of MDR/RR-TB. Each dot represents one of the 200 simulation runs. Ellipses indicate the 95% confidence regions for simulation results, summarising the range and uncertainty of the ICER estimates. There is a 95% probability that the true incremental cost and effect (and hence the ICER) lie within the area enclosed by the ellipse. ICER Incremental cost-effectiveness ratio; BPaL bedaquiline, pretomanid and linezolid; BPaLM bedaquiline, pretomanid, linezolid and moxifloxacin; MDR/RR-TB multidrug- or rifampicin-resistant tuberculosis.

prevalence (0%-90%), moxifloxacin prices ($0.1-$1.0), and difference in risk ratios between BPaLM and BPaL compared to traditional regimens (0.01-0.22), while keeping all other parameters fixed at baseline values (as specified in Table 1 and S2 Table) (Fig 3). We selected these ranges to test the robustness of our results. While our choice of the lower and upper bounds extend beyond typical observed values in some cases (e.g., 90% fluoroquinolone resistance), these allow for exploring extreme scenarios, which can inform policy decisions under highly varying conditions. Furthermore, we explored the interactions between these parameters by analysing how the impact of fluoroquinolone resistance varied under different moxifloxacin prices and the difference in risk ratios between BPaLM and BPaL compared to traditional regimens (S2, S3 Figs).

## Results

In the absence of fluoroquinolone DST, empirical use of BPaLM as a potential replacement for BPaL had an incremental cost-effectiveness ratio of $58 (interquartile range: $49-$73), $32 (IQR: $23-$53), $35 (IQR: $28 - $51), $174 (IQR: $161 - $209) per DALY averted in Georgia, India, the Philippines, and South Africa respectively (**Table 2** and Fig 1). The incremental costs per person of BPaLM compared with BPaL were $77, $30, $50 and $184, with higher DALYs averted per person of 1.3, 1.0, 1.4 and 1.1 in Georgia, India, the Philippines and South Africa, respectively. Probabilistic sensitivity analysis (Fig 1) also suggests that BPaLM consistently produces higher costs and better health outcomes than BPaL in all simulations, though there is some uncertainty in both costs and health outcomes as indicated by confidence ellipses that extend beyond this quadrant. The breakdown of costs in S4 Table shows that variations in ICERs between countries reflect differences in underlying healthcare cost components across these settings.

**Table 1. Model parameters.**

| Parameter | Baseline value | Standard Error | PSA Distribution | Reference |
|---|---|---|---|---|
| Updated parameters | | | | |
| Fluoroquinolone resistance prevalence | | | | |
| *Georgia* | 27% | | N/A | 2019 data in [15] |
| *India* | 37% | | N/A | 2019 data in [15] |
| *Philippines* | 9% | | N/A | 2019 data in [15] |
| *South Africa* | 19% | | N/A | 2019 data in [15] |
| Estimated risk ratio for treatment success | | | | |
| BPaL compared to traditional regimen (RR_BPaL) | 1.03* | 0.05 | Log normal | Estimated based on [11] and [15] [Equation 1, 2] |
| BPaLM compared to traditional regimen (RR_BPaLM) | 1.15* | 0.06 | Log normal | Estimated based on [11] and [15] [Equation 1, 2] |
| *Treatment success of traditional regimen* | | | | |
| *Georgia* | 69% | | N/A | 2019 data in [34] |
| *India* | 58% | | N/A | 2019 data in [34] |
| *Philippines* | 68% | | N/A | 2019 data in [34] |
| *South Africa* | 65% | | N/A | 2019 data in [34] |

*The estimated actual risk ratio that compares the likelihood of achieving a favourable outcome (no unfavourable outcome) with the BPaLM regimen compared to the traditional regimen. This has been estimated using [Equation 1] and [Equation 2].

BPaL bedaquiline, pretomanid and linezolid; BPaLM bedaquiline, pretomanid, linezolid and moxifloxacin; PSA probabilistic sensitivity analysis.

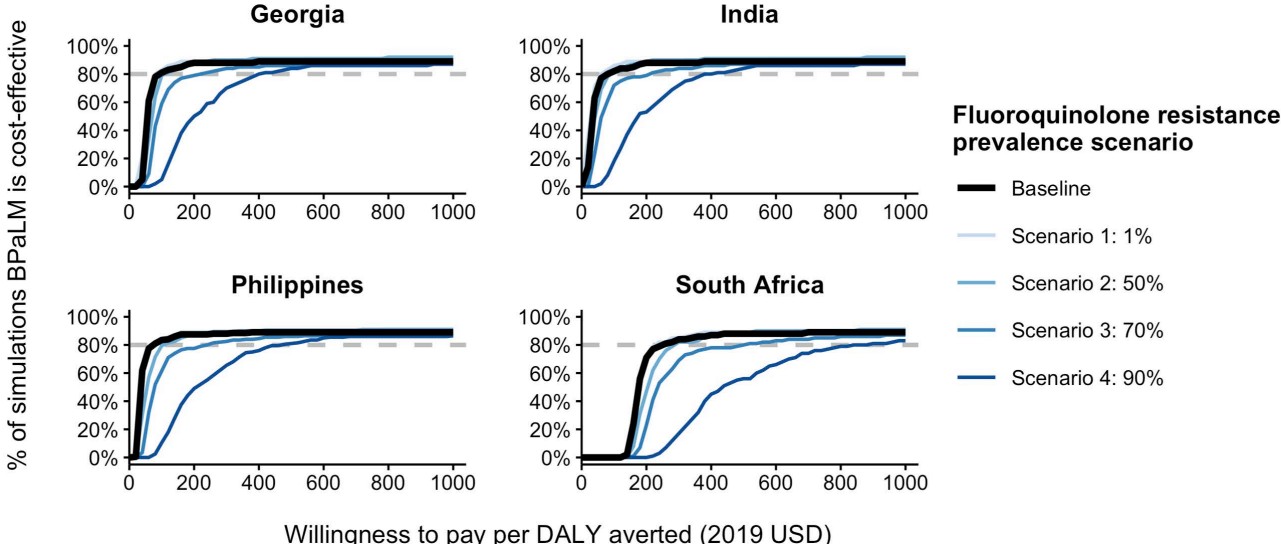

**Fig 2. Cost-effectiveness acceptability curves by fluoroquinolone resistance prevalence.** This figure shows the cost-effectiveness acceptability curves for the empirical use of the BPaLM regimen as a replacement for the BPaL regimen under varying levels of fluoroquinolone resistance prevalence. Each line represents a different scenario of fluoroquinolone resistance prevalence explored in the probabilistic sensitivity analysis. The x-axis shows the willingness-to-pay per disability-adjusted life year averted, while the y-axis indicates the probability that the BPaLM regimen is a cost-effective alternative to the BPaL regimen at the given willingness to pay. Country-specific WTP thresholds are as follows: $2,419 for Georgia, $422 for India, $962 for the Philippines, and $3,951 for South Africa (all in USD) [21]. BPaL bedaquiline, pretomanid and linezolid; BPaLM bedaquiline, pretomanid, linezolid and moxifloxacin. * The irregularities in the curves reflect the limited number of sampling iterations (200 runs) in the probabilistic sensitivity analysis.

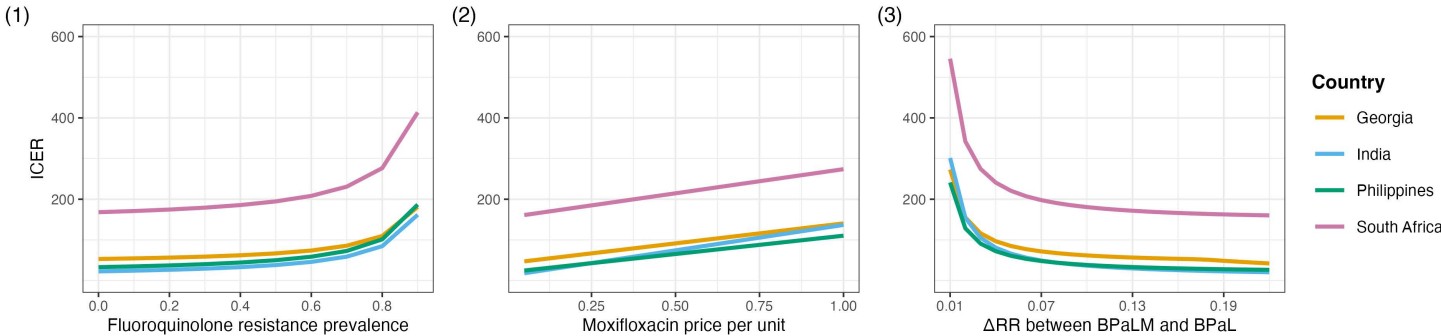

**Fig 3. Univariate sensitivity analyses comparing BPaLM and BPaL regimens in four countries for varying only [1] fluoroquinolone resistance prevalence, [2] moxifloxacin unit cost per pill, and [3] differences in risk ratios between BPaLM and BPaL compared to traditional regimens (where the BPaL risk ratio relative to the traditional regimens is 1.03). All other parameters are kept at baseline levels (Table 1 and S2 Table). BPaL bedaquiline, pretomanid and linezolid; BPaLM bedaquiline, pretomanid, linezolid and moxifloxacin.**

**Table 2. ICER in empirical treatment scenario.** Incremental cost per additional disability-adjusted life years averted by using the BPaLM regimen instead of the BPaL regimen in an empirical treatment setting without fluoroquinolone drug susceptibility testing. Interquartile ranges reflect the uncertainty around the ICER estimates based on probabilistic sensitivity analyses. DALYs disability-adjusted life years; ICER Incremental cost-effectiveness ratio.

| Country | Regimen | Total costs per person ($2019) | Total DALYs per person | Comparison of BPaLM replacing BPaL | | | |
|---|---|---|---|---|---|---|---|
| | | | | Incremental costs per person ($2019) | Incremental DALYs averted per person | ICER ($2019) | Interquartile range ($2019) |
| **Georgia** | BPaL | $3,236 | 6.8 | | | | |
| | BPaLM | $3,313 | 5.4 | 77 | 1.3 | 58 | 49-73 |
| **India** | BPaL | $869 | 10.1 | | | | |
| | BPaLM | $899 | 9.1 | 30 | 1.0 | 32 | 23-53 |
| **Philippines** | BPaL | $1,083 | 8.2 | | | | |
| | BPaLM | $1,132 | 6.8 | 50 | 1.4 | 35 | 28-51 |
| **South Africa** | BPaL | $3,380 | 10.1 | | | | |
| | BPaLM | $3,564 | 9.1 | 184 | 1.1 | 174 | 161-209 |

Cost-effectiveness acceptability curves (Fig 2) indicate that BPaLM has a high probability of being cost-effective compared to BPaL across varying levels of fluoroquinolone resistance. In most scenarios, the probability of cost-effectiveness exceeds 80% even before the willingness-to-pay reaches the country-specific cost-effectiveness thresholds (Georgia $2,419; India $422; the Philippines $962; South Africa $3,951 per DALY averted in USD) [21]. This certainty decreased with higher levels of fluoroquinolone resistance, but this was only most significant when prevalence of fluoroquinolone resistance reached 90%.

Sensitivity analyses revealed that key influential parameters for the cost-effectiveness were variable across countries. Fig 3 shows three different relationships. ICERs increase exponentially with fluoroquinolone resistance, particularly above 70%, while they initially decline steeply and then flatten around 0.07 as the difference in risk ratios between BPaLM and BPaL compared to traditional regimens increase. ICER values increase linearly with moxifloxacin prices. The interaction between these parameters (S2, S3 Figs) showed that both higher moxifloxacin prices and lower differences in risk ratios between BPaLM and BPaL compared to traditional regimens amplified the effect of fluoroquinolone resistance on ICERs. Higher moxifloxacin prices increased ICERs at high resistance levels, while lower differences in risk ratios between BPaLM and BPaL compared to traditional regimens led to steeper ICER increases.

S4 Fig illustrates comprehensive results from our one-way sensitivity analysis using a tornado diagram. In Georgia, non-drug costs had the largest impact on ICERs, while in India, where total treatment costs were lower, the impact of these parameters on ICERs was more modest. The cost-effectiveness plane (S5 Fig) confirmed that two key parameters consistently drove outcome variation across settings, with fluoroquinolone resistance prevalence and the difference in risk ratios between BPaLM and BPaL compared with the traditional regimens being the main drivers. In particular, Fig 1 shows a linear trend between cost and DALYs across all settings. S5 Fig suggests that this linearity may have been driven by the difference in risk ratios between BPaLM and BPaL compared with the traditional regimens.

## Discussion

Our results suggest that the BPaLM regimen could serve as a cost-effective replacement for the BPaL regimen in empirical treatment settings across the four study countries. While BPaLM incurred higher costs than BPaL, it also averted more DALYs, with the incremental cost-effectiveness ratios falling below cost-effectiveness thresholds [21]. This finding remained robust across a feasible fluoroquinolone resistance prevalence range. The benefits of including moxifloxacin in empirical treatment outweigh the potential harms, even in settings with relatively high fluoroquinolone resistance levels of 20–40% [15]. The cost-effectiveness remained favourable until the fluoroquinolone resistance prevalence reached extremely high levels (> 90%), which are currently not observed in any setting.

Our analysis addresses a critical gap in the literature by directly comparing the cost-effectiveness of BPaLM versus BPaL regimens in empirical treatment settings across varying fluoroquinolone resistance prevalences. Recent work [17] employed a quantitative approach to capture the trade-off between efficacy and safety of regimens in empirical treatment settings at different fluoroquinolone resistance prevalences. However, this did not consider the new short all-oral regimens. A previous analysis found that BPaL was the most cost-saving option, while BPaLM was preferred at a willingness-to-pay threshold of 0.5 × GDP per disability-adjusted life year averted [12]. More recently, a separate analysis using data collected directly from the TB PRACTECAL trial demonstrated the cost-saving potential from a societal perspective of BPaL-based regimens in Belarus, Uzbekistan and South Africa [22]. While the available evidence consistently suggests that BPaL-based regimens offer advantages over traditional regimens, these studies did not address the critical question of which regimen to choose in empirical treatment settings. In addition, James et al. [18] compared BPaLM with traditional regimens and examined options for switching regimens if a patient had to discontinue moxifloxacin due to side effects or resistance acquisition. While this analysis considered treatment modifications, it did not directly compare BPaLM with BPaL as initial empirical treatment options, nor did it evaluate cost-effectiveness across varying fluoroquinolone resistance prevalences. Our study addresses this specific gap by providing the first economic comparison of these two WHO-recommended regimens under empirical treatment conditions, accounting for the trade-offs when DST results are unavailable and treatment decisions must be made based on population-level resistance patterns.

The varying incremental cost-effectiveness ratios across countries reflect several interconnected factors, including underlying healthcare cost components, fluoroquinolone resistance prevalence, and health system infrastructure. Countries with higher healthcare service costs, such as South Africa and Georgia, showed correspondingly higher ICERs. Differences in care delivery models, such as the requirement in Georgia for inpatient treatment periods compared to primarily outpatient care elsewhere, may have also contributed to these cost variations. In addition, baseline treatment success rates varied across settings, affecting the potential for improvement with new regimens. Despite these diverse epidemiological, clinical, and health system contexts, the consistent cost-effectiveness of BPaLM as a replacement for BPaL suggests that our findings may be generalisable to similar settings.

Our study should be interpreted with caution. First, this analysis used fluoroquinolone resistance as a proxy for moxifloxacin resistance, and assumed that fluoroquinolone resistance completely negates any treatment benefit from including moxifloxacin in the regimen. However, some fluoroquinolone-resistant mutants may retain partial susceptibility to

moxifloxacin [23,24]. This suggests that our results are conservative regarding the efficacy of BPaLM, which reinforces our main findings. Our analysis relies on modelled estimates and extrapolated data due to limited real-world evidence on BPaLM effectiveness in routine programmatic settings. Our treatment success estimates were derived from back-calculated risk ratios rather than directly observed country-specific data, where resistance status is often unknown, introducing uncertainty around real-world effectiveness based on resistance status. Second, the interpretation of this study should be limited to the empirical treatment setting. While rapid access to comprehensive drug susceptibility testing remains the optimal approach, our analysis did not compare the empirical treatment scenario with a scenario where patients receive regimens based on individual DST results. Therefore, we specifically excluded all costs associated with second-line drug susceptibility testing from our analysis (including both liquid culture and molecular testing methods). This approach allows our findings to be directly applicable to situations where rapid DST results are not yet available. Third, an important consideration not assessed in our analysis is the potential impact of widespread empiric BPaLM use on the development of broader fluoroquinolone resistance. Since fluoroquinolones are used to treat various conditions beyond tuberculosis, their widespread empiric use could contribute to resistance without proper stewardship measures, thereby compromising treatment options for other infections [25,26]. Finally, treatment outcomes other than success (such as loss to follow-up, treatment failure, and death) were not directly adjusted for fluoroquinolone resistance status in our model. This simplification could affect our DALY calculations, as patients with fluoroquinolone-resistant TB may have different rates compared to those with fluoroquinolone-susceptible TB. For instance, those with resistance may experience higher treatment failure rates because fluoroquinolones are important components in MDR/RR-TB regimens [27]. In addition, all patients who failed their initial treatment regimen were switched to a long SOC 'rescue' regimen, regardless of the initial treatment received. However, we did not explicitly model adverse event-related regimen changes separately due to limited data on switching patterns in empirical settings. This simplified approach may not fully capture the complexity of real-world treatments.

A key area for further research is the impact of resistance patterns beyond fluoroquinolones, such as bedaquiline resistance, which has already been identified in multiple countries [28–32]. Although our sensitivity analyses varying the relative treatment effect sizes and drug costs helped explore this uncertainty, the emergence of resistance to other key drugs in BPaL-based regimens could significantly affect their efficacy, necessitating continued monitoring and re-evaluation of empirical treatment strategies. Furthermore, it is important to evaluate empirical treatment choices for other regimens [33].

Our findings support the empirical use of BPaLM as a potential replacement for BPaL for the treatment of MDR/RR-TB in the absence of fluoroquinolone DST. These results build on previous cost-effectiveness analyses by explicitly evaluating empirical treatment scenarios in settings with different fluoroquinolone resistance patterns, an important real-world consideration as countries scale up new MDR/RR-TB regimens with limited access to rapid DST. Our analytical framework for evaluating empirical treatment strategies could inform similar policy decisions for other infectious diseases where drug resistance and diagnostic constraints are important considerations. The empirical use of BPaLM remains favourable even if fluoroquinolone resistance prevalence were to increase, reinforcing recent WHO recommendations [6].

## Supporting information

**S1 Checklist. CHEERS 2022 checklist.**
(DOCX)

**S1 Text. Supplementary methods.**
(DOCX)

**S2 Text. Updated efficacy model.**
(DOCX)

**S1 Table. Treatment scenarios.** BPaL bedaquiline, pretomanid and linezolid; BPaLM bedaquiline, pretomanid, linezolid and moxifloxacin; MDR/RR-TB multidrug- or rifampicin-resistant tuberculosis; pre-XDR-TB pre-extensively drug-resistant TB.
(DOCX)

**S2 Table. Country-specific and non-specific parameters.**
(DOCX)

**S3 Table. Parameter ranges used in univariate sensitivity analyses.**
(DOCX)

**S4 Table. Breakdown of per-person costs by treatment regimen and country.**
(DOCX)

**S1 Fig. Model structure.** The model structure is based on Sweeney et al. [12]. BPaL, bedaquiline, pretomanid, and linezolid; SOC, standard of care; TB, tuberculosis.
(TIF)

**S2 Fig. Assessing the impact of other variables on the impact of fluoroquinolone resistance on ICER: moxifloxacin unit prices (baseline = 0.16).** ICER Incremental cost-effectiveness ratio; BPaL bedaquiline, pretomanid and linezolid; BPaLM bedaquiline, pretomanid, linezolid and moxifloxacin; ΔRR, difference in risk ratios.
(TIF)

**S3 Fig. Assessing the impact of other variables on the impact of fluoroquinolone resistance on ICER: differences in risk ratios between BPaLM and BPaL compared to traditional regimens (BPaL risk ratio relative to the traditional regimens = 1.03).** ICER Incremental cost-effectiveness ratio; BPaL bedaquiline, pretomanid and linezolid; BPaLM bedaquiline, pretomanid, linezolid and moxifloxacin; ΔRR, difference in risk ratios.
(TIF)

**S4 Fig. Tornado diagrams showing the effect of individually varying key parameters on the ICER of BPaLM as a replacement for BPaL.** Bars represent the absolute change in ICER (relative to baseline) for high and low values of parameters with all other parameters at baseline values. AE, adverse event; BPaL, bedaquiline, pretomanid and linezolid; BPaLM, bedaquiline, pretomanid, linezolid and moxifloxacin; FQ, fluoroquinolone; ICER, Incremental cost-effectiveness ratio; LTFU, lost to follow-up; ΔRR, difference in risk ratios.
(TIF)

**S5 Fig. Cost-effectiveness plane showing the effect of individually varying key parameters on the ICER of BPaLM as a replacement for BPaL.** Each line represents the impact of varying a single parameter with all other parameters at baseline values. AE, adverse event; BPaL, bedaquiline, pretomanid and linezolid; BPaLM, bedaquiline, pretomanid, linezolid and moxifloxacin; FQ, fluoroquinolone; ICER, Incremental cost-effectiveness ratio; LTFU, lost to follow-up; ΔRR, difference in risk ratios.
(TIF)

## Acknowledgments

We sincerely thank Dr. Bern-Thomas Nyang'wa, Dr. Matthew Dodd, Martin Harker, Naomi Fuller, and Dr. Simon Procter for their helpful discussions. The views expressed in this manuscript do not necessarily reflect the views of their affiliated institutions or funding bodies.

## Author contributions

**Conceptualization:** Chaelin Kim, Gwenan M Knight, C. Finn McQuaid.

**Data curation:** Sedona Sweeney.

**Formal analysis:** Chaelin Kim.

**Funding acquisition:** Chaelin Kim, Gwenan M Knight, C. Finn McQuaid.

**Investigation:** Chaelin Kim, Gwenan M Knight, C. Finn McQuaid.

**Methodology:** Chaelin Kim, Sedona Sweeney, Hojoon Sohn, Gwenan M Knight, C. Finn McQuaid.

**Project administration:** Chaelin Kim.

**Resources:** Sedona Sweeney.

**Software:** Chaelin Kim.

**Supervision:** Hojoon Sohn, Gwenan M Knight, C. Finn McQuaid.

**Validation:** Sedona Sweeney.

**Visualization:** Chaelin Kim.

**Writing – original draft:** Chaelin Kim.

**Writing – review & editing:** Chaelin Kim, Sedona Sweeney, Hojoon Sohn, Gwenan M Knight, C. Finn McQuaid.

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
