## [Decision Letter · Decision Letter 0]

12 Aug 2025

PGPH-D-25-00750

Impact of fluoroquinolone resistance on the cost-effectiveness of empiric treatment for multidrug- or rifampicin-resistant tuberculosis

Dear Dr. Kim,

Thank you for submitting your manuscript to PLOS Global Public Health. After careful consideration, we feel that it has merit but does not fully meet PLOS Global Public Health’s publication criteria as it currently stands. Therefore, we invite you to submit a revised version of the manuscript that addresses the points raised during the review process.

We look forward to receiving your revised manuscript.

Kind regards,

Stefan Kohler, Ph.D., M.D.

Academic Editor

Journal Requirements:

1. Please ensure you have included the registration number for the clinical trial referenced in the manuscript.

1. Please clarify all sources of funding (financial or material support) for your study. List the grants (with grant number) or organizations (with url) that supported your study, including funding received from your institution. 

2. State the initials, alongside each funding source, of each author to receive each grant.

3. State what role the funders took in the study. If the funders had no role in your study, please state: “The funders had no role in study design, data collection and analysis, decision to publish, or preparation of the manuscript.”

4. If any authors received a salary from any of your funders, please state which authors and which funders.

3. Please provide separate figure files in .tif or .eps format.

4. Your manuscript is missing the following sections: Abstract. Please ensure these are present, and in the correct order, and that any references to subheadings in your main text are correct. An outline of the required sections can be consulted in our submission guidelines here: 

https://journals.plos.org/globalpublichealth/s/submission-guidelines#loc-parts-of-a-submission

5. We have noticed that you have uploaded Supporting Information files, but you have not included a list of legends. Please add a full list of legends for your Supporting Information files after the references list. 

Reviewers' comments:

Reviewer's Responses to Questions

**Comments to the Author**

1. Does this manuscript meet PLOS Global Public Health’s publication criteria ? Is the manuscript technically sound, and do the data support the conclusions? The manuscript must describe methodologically and ethically rigorous research with conclusions that are appropriately drawn based on the data presented.

Reviewer #1: Partly

Reviewer #2: Partly

Reviewer #3: Yes

2. Has the statistical analysis been performed appropriately and rigorously?

Reviewer #1: Yes

Reviewer #2: Yes

Reviewer #3: I don't know

3. Have the authors made all data underlying the findings in their manuscript fully available (please refer to the Data Availability Statement at the start of the manuscript PDF file)?

Reviewer #1: Yes

Reviewer #2: Yes

Reviewer #3: Yes

4. Is the manuscript presented in an intelligible fashion and written in standard English?

Reviewer #1: Yes

Reviewer #2: Yes

Reviewer #3: Yes

5. Review Comments to the Author

Reviewer #1: The document is well written, but my initial impression is that it is more an extension of previous work than new results of public interest. I would suggest that the authors review the introduction to this paper to avoid this perception. Nevertheless, the question of whether it is necessary to introduce a new drug into the treatment scheme for TB should be discussed and studied further.

It is essential to follow the CHEERS statement to clearly state the study perspective, materials and methods, among other things. Given the uncertainty surrounding the parameters used in this exercise, sensitivity analysis should be expanded accordingly.

As health economics studies tend to rely more on hypothetical than real data, this should be discussed more in the paper.

In addition, the differences between the four countries' health systems should be further elaborated on to give the reader a better understanding of the results and recommendations.

Reviewer #2: The authors addressed a significant and timely issue. They estimated the cost-effectiveness of the BPaLM regimen with BPal in the absence of DST for fluoroquinolone. The methods and techniques they utilized are in line with the standards in the literature.

I have one major and several minor comments.

They indicated one disadvantage of BPaLM over BPAL as more adverse events (Line 60). However, they have not used this information in their analysis. (At least they do not explicitly explain how they used it) Presumably, more adverse events would increase the costs or impact health outcomes or both thus estimated cost effectiveness. Ideally, they could explicitly incorporate these into the decision framework. Since they model the empirical treatment, the decision analytical framework could include the option for physicians to switch the BPaL regime in case of a significant adverse event. If that is not possible due to data availability, some discussion of the issue is warranted in the discussion and limitations sections.

Minor Comments:

• They used 50% country's GDP as the threshold for willingness to pay. Why don’t they use the country-specific estimates published in the literature (Woods et al (2016)1, or Riviere et al (2023)2 …)

• They used 2019 GDP values; it would have been better if they used more recent GDP levels.

• In Line 70, prevalence rates of fluoroquinolone-resistant TB are given. It is not clear what exactly these rates are. (Among TB patients, among DR-TB, etc?)

• In line 128, the authors indicated that “We assumed that the efficacy of BPaLM in the pre-XDR-TB population would be the same as that of BPaL”. What is the advantage of this assumption? If I understand equations 1 and 2, that is not necessary.

• In Table 1, RR_BPaLM is smaller than RR_BPaL. Is this a typo? If not, I do not understand that.

• In Figure 2, including the willingness to pay threshold would make it easier for the audience to comprehend the cost-effectiveness.

• The authors might explain more elaborately the additional value of their study over the reference 12 and probably reference 18. In order to do that, some discussion about the reasons for the empirical treatment cost of DST for fluoroquinolone would be useful.

• The references to the links in the PDF file go to a lean library account. This makes it difficult to read for non-account holders.

1. Woods B, Revill P, Sculpher M, Claxton K. Country-Level Cost-Effectiveness Thresholds: Initial Estimates and the Need for Further Research. Value Health. 2016;19(8):929-935. doi:10.1016/j.jval.2016.02.017

2. Pichon-Riviere A, Drummond M, Palacios A, Garcia-Marti S, Augustovski F. Determining the efficiency path to universal health coverage: cost-effectiveness thresholds for 174 countries based on growth in life expectancy and health expenditures. Lancet Glob Health. 2023;11(6):e833-e842. doi:10.1016/S2214-109X(23)00162-6

Reviewer #3: Impact of fluoroquinolone resistance on the cost-effectiveness of empiric treatment for multidrug- or rifampicin-resistant tuberculosis

Overview

Thank you for the opportunity to review this manuscript which evaluates cost-effectiveness of BPaLM) compared to BPaL for the empiric treatment of MDR/RR-TB. This is an important study that evaluates realistic scenarios and has potential to directly inform policy and practice in the context of limited or absence of FQ DST.

Comments

What is the potential impact of empiric treatment on wider resistance to fluoroquinolones? Even if this was not evaluated, perhaps it’s important to discuss that considering these drugs are used for other conditions and if widespread resistance develops, this would reduce therapeutic options in limited resource settings.

Was this analysis prespecified in a HEAP? This is increasingly being considered good practice.

How do 0.5X per capita GDP thresholds compare with opportunity-cost based thresholds published over the past few years? While I think this choice is pragmatic, the absolute values seem large and I wondered if you are able to comment on how the results would be affected by a lower CET of less than 500 USD which is commonly suggested in most LMIC with high TB burden.

6. PLOS authors have the option to publish the peer review history of their article (what does this mean? ). If published, this will include your full peer review and any attached files.

**Do you want your identity to be public for this peer review?** For information about this choice, including consent withdrawal, please see our Privacy Policy .

Reviewer #1: No

Reviewer #2: No

Reviewer #3: **Yes: ** Nyashadzaishe Mafirakureva

---

## [Editor Report · Decision Letter 1]

16 Sep 2025

Impact of fluoroquinolone resistance on the cost-effectiveness of empiric treatment for multidrug- or rifampicin-resistant tuberculosis

PGPH-D-25-00750R1

Dear Miss Kim,

We are pleased to inform you that your manuscript 'Impact of fluoroquinolone resistance on the cost-effectiveness of empiric treatment for multidrug- or rifampicin-resistant tuberculosis' has been provisionally accepted for publication in PLOS Global Public Health.

Best regards,

Stefan Kohler, Ph.D., M.D.

Academic Editor